# The Effect of Sponge City Construction for Reducing Directly Connected Impervious Areas on Hydrological Responses at the Urban Catchment Scale

**Changmei Liang [1,2,3], Xiang Zhang [1,2,3,\*], Jun Xia [1,2,3], Jing Xu [1,2,3,\*] and Dunxian She [1,2,3]**

1    State Key Laboratory of Water Resources and Hydropower Engineering Science, Wuhan University,
     Wuhan 430072, China; liangchangmei@whu.edu.cn (C.L.); xiajun666@whu.edu.cn (J.X.);
     shedunxian@whu.edu.cn (D.S.)
2    Hubei Provincial Key Lab of Water System Science for Sponge City Construction, Wuhan 430072, China
3    Hubei Provincial Collaborative Innovation Center for Water Resources Security, Wuhan 430072, China
\*    Correspondence: zhangxiang@whu.edu.cn (X.Z.); jingxu0506@whu.edu.cn (J.X.)

**Abstract:** Low-impact development (LID) has been widely used at both site-specific and local scales to try and mitigate the impact of urban stormwater runoff caused by increasing impervious urban areas. Recently, the concept of a "sponge city" was proposed by the Chinese government, which includes LID controls at the source, a pipe drainage system midway, and a drainage system for excess stormwater at the terminal. There is a need to evaluate the effectiveness of sponge city construction at the large urban catchment scale, particularly with different spatial distributions of LIDs that reduce directly connected impervious areas (DCIAs). In this paper, the performances of five design scenarios with different spatial distributions but same sizes of LID controls at the urban catchment scale were analyzed using a geographic information system (GIS) of the United States Environmental Systems Research Institute (ESRI)—based Storm Water Management Model (SWMM) of the United States Environmental Protection Agency (USEPA) and MIKE 11 of Danish Hydraulic Institute (DHI) in Xining City, China. Results confirmed the effectiveness of sponge city construction in reducing the urban stormwater runoff. The hydrological performance reduction was positively correlated and linearly dependent on DCIA reduction. Peak flow reduction was most sensitive to DCIA reduction, followed by runoff volume and peak time. As rainfall intensity increased, the hydrological performance was more sensitive to rainfall intensity than DCIA reduction. Results of this study provide new insights for stormwater managers to implement LID more effectively at the urban catchment scale.

**Keywords:** sponge city construction; stormwater management; directly connected impervious area; urban catchment scale

---

## 1. Introduction

With rapid economic developments, global urbanization has increased continuously, especially in China. Large areas of grassland and farmland were converted into roads and buildings, resulting in an increase in impervious areas. These changes directly lead to changes in the previous infiltration processes, thereby altering the natural hydrological water cycle. The changes to the surface runoff characteristics of these areas include larger runoff volume, higher peak runoff, earlier peak runoff time, and poorer water quality [1–4]. Of these impacts, urban flood disasters have appeared more frequently, which increasingly threatens people's lives and health, resulting in huge economic losses.

To mitigate these impacts, a new concept in stormwater management strategies was announced by the Chinese government in 2013, called a "sponge city" [5]. Deviating from the conventional

"rapid-draining" approach, this new paradigm called for the use of natural processes as part of the urban runoff control strategy. During 2015 and 2016, a total of 30 cities were selected as pilot cities to implement the sponge city in China [5–7]. Many studies about sponge city construction in China have evaluated the hydrological effects of low-impact development (LID) controls implementation. For example, Qin et al. [8] analyzed the effect of LID control under different rainfall design scenarios on urban flooding at the urban drainage system scale. W. Liu et al. [9] developed a simple model to quantify the effectiveness of green infrastructure on reducing the volume and peak flow of urban flooding at the community scale. Kong et al. [10] explored the hydrological performance of LID control under four different land-use scenarios at the city scale. Previous studies that assessed the effectiveness of LID controls were mostly on a small scale. Sponge city construction is at a larger scale, which includes LID implementation at the source, a drainage pipe system at the midway, and an excessive stormwater drainage system, such as rivers and lakes, at the terminal [5,11]. A limited number of studies have evaluated the effectiveness of sponge city construction from the large urban catchment scale that includes source, midway, and terminal assessment.

Modeling the effectiveness of sponge city construction is necessary for decision makers to generalize best construction practices and provide guidance for stormwater management [10]. Hydrological models are often used to simulate runoff and pollutant loads, which assess the effectiveness of LID controls at various temporal–spatial scales [12–14]. There are many kinds of distributed hydrological models, such as the System for Urban Stormwater Treatment and Analysis Integration (SUSTAIN) model of the United States Environmental Protection Agency (USEPA), the Hydrological Simulation Program–Fortran (HSPF) model of USEPA, the Soil Water Assessment Tool (SWAT) of the United States Department of Agriculture (USDA), the Model for Urban Stormwater Improvement Conceptualization (MUSIC) of Australian Department of Water Service, and the Storm Water Management Model (SWMM) of USEPA. The SWMM has been widely used and has proven to be helpful in large urban catchment planning [15–17]. However, to simulate large-scale sponge city construction, it is challenging for SWMM to account for the excessive stormwater in drainage systems at the terminal, like rivers and lakes. To address this issue, hydrodynamic models are used to simulate the water flow in rivers at the terminal. MIKE 11 is a simulation system with a clear structure, a friendly interface, and a wide range of applications. It can be widely used in many research areas, such as simulating water flows in estuaries and river networks, regional analysis of drainage systems, and optimization of ground drainage schemes [18–21].

Total impervious area (TIA) is an important indicator of urbanization and is used to describe the effect of urbanization on stormwater systems [22]. However, recent studies [10,23] showed that the TIA may not be an accurate parameter for estimating runoff, because not all stormwater runoff from the TIA flows into the downstream stormwater pipes. Therefore, directly connected impervious area (DCIA) has been proposed as an alternative metric [24,25]. DCIA is a fraction of the impervious area that is hydraulically connected to downstream drainage by a buried pipe route. Previous studies have demonstrated that DCIA is more dependable than TIA [26–28]. Reducing the DCIA can significantly reduce the peak runoff and runoff volume and can delay the peak runoff time [26–28]. Many earlier studies developed methods to quantify the DCIA [23,26,28]. For example, William M. Alley [29] developed an empirical relationship between TIA and DCIA based on area of different land uses in 19 urban basins in Metropolitan Denver. Ebrahimian et al. [28] quantified the DCIA in urban catchments by using an improved rainfall–runoff data analysis method to eliminate the subjective part of the existing method and reduce the uncertainty of effective impervious area (EIA) estimates. However, these methods have been demonstrated to be variable and lack the spatial distribution considerations of DCIA [23,26]. By using detailed geographic information system (GIS) and database tools which combine the sewer system and land use, the spatial distribution of DCIA can be identified and the accuracy of the estimated runoff can be increased [26,27,30]. The application of LID controls, especially if the spatial distribution is considered, can reduce DCIA significantly [10]. However, there

are few studies concerning how to optimize the spatial distribution of LID controls while considering DCIA reduction.

In this study, a framework was constructed using the GIS-based SWMM and MIKE 11 to simulate stormwater runoff at the large urban region scale. Focusing on a case study in Xining, Qinhai province, China, the condition of local land was collected at the study area, including soil experiments, rainfall analysis, land-use classification, topology, pipe drainage system, and information about cross-sections of the river. Based on these basic data, the distribution of DCIA was identified by GIS, and five design scenarios (i.e., S1–S5) with different spatial distributions but the same sizes of rain garden, permeable pavement, and vegetated swale were determined. These scenarios reduced the DCIA by 0% to 100% for three different return periods (T = 2, 5, and 10 years) which were set. The hydrological performance (peak flow, runoff volume, and peak time) of these scenarios were simulated by SWMM and MIKE 11 and were compared (Figure 1). The main objectives of this paper were to: (1) investigate the effectiveness of sponge city construction at a large urban regional scale at the source, midway, and terminal, (2) evaluate the potential hydrological response from the implementation of LID controls under different spatial distributions with reduced DCIA. The results of this paper can provide technical guidance for the implementation of LID controls and can support improved decision making in stormwater management and planning.

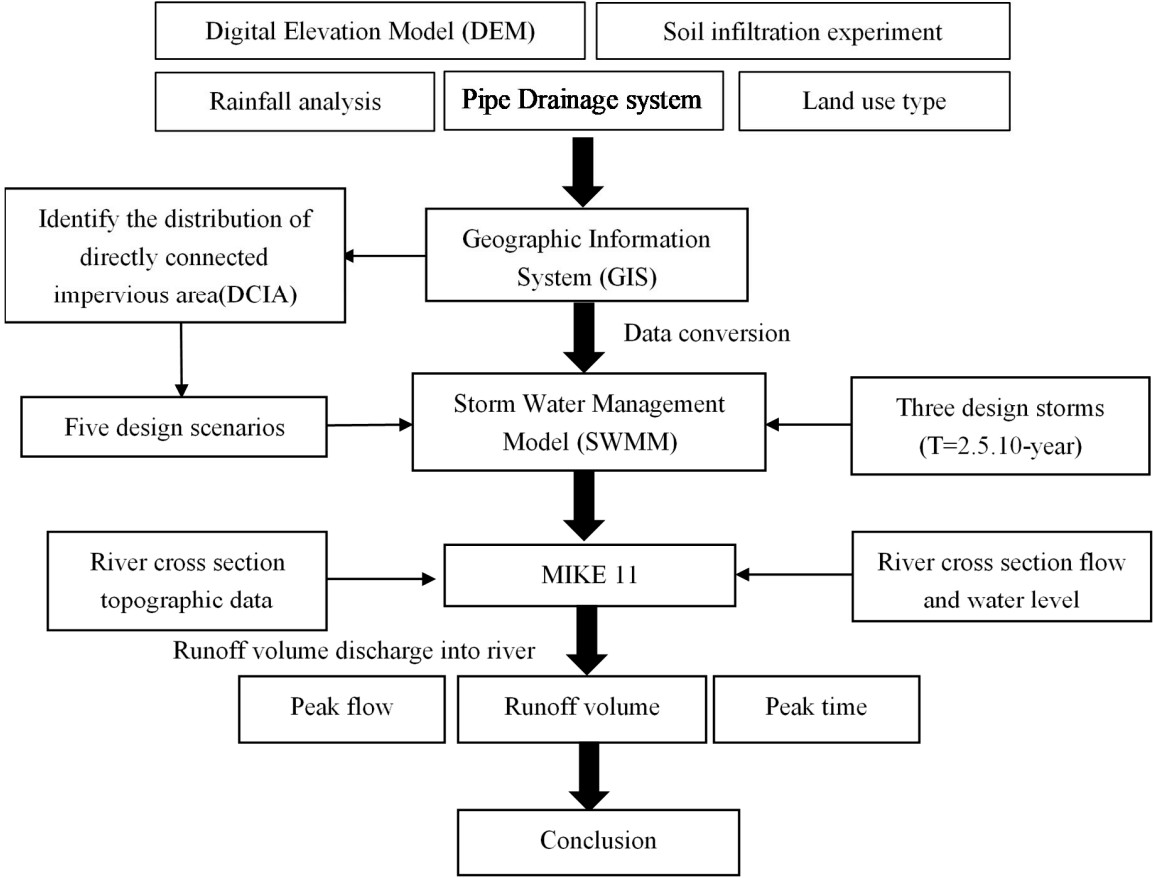

**Figure 1.** Description of the data requirements and processes.

## 2. Materials and Methods

### 2.1. Study Area

Xining (36°62′ N, 101°77′ E) is a city located in Qinghai Province, northwest China, as shown in Figure 2a. It is surrounded by mountains. Its climate is semi-arid. The average annual temperature is

7.6 °C. The average annual rainfall is 410 mm with 90% of the rainfall occurring between April and August. The study area in Xining City is in an economic development zone with an area of 21.2 km$^2$. It is located at the eastern side of Xining. The study area was classified into six different land-use types: road, greenspace, pavement, roofs, farmland, parking. The impervious areas, such as roads, car parks, and roofs, occupy more than 30% of the total area (Table 1). The 8.2 km long Huangshui River is located in the study area and the water from the pipe network discharges directly into it.

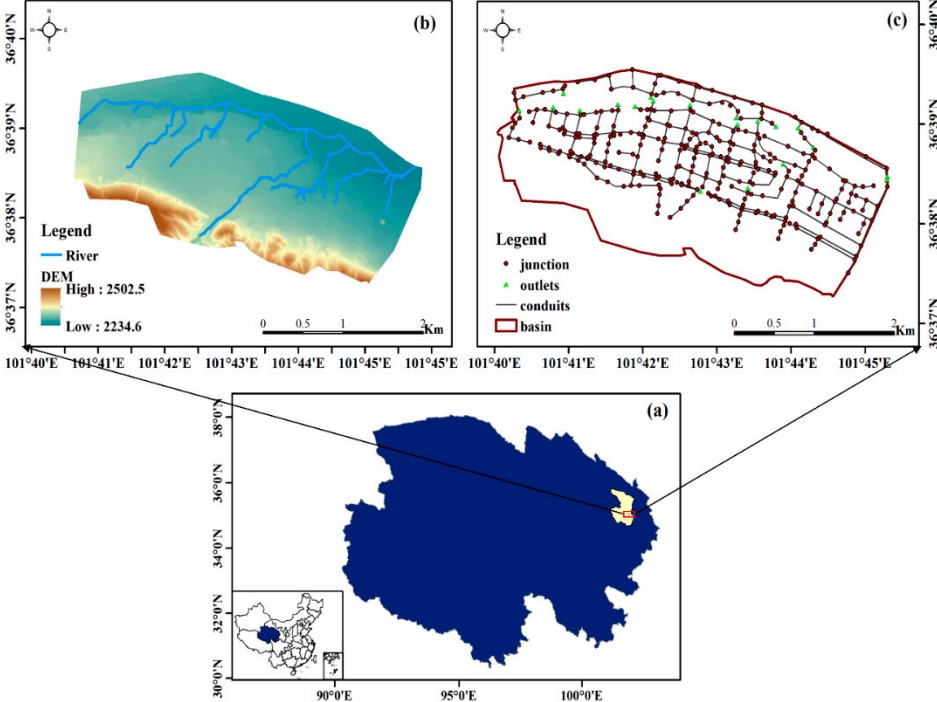

**Figure 2.** Maps of study area: (**a**) location of Xining and study area in China; (**b**) DEM (Digital Elevation Model) of study area including natural rivers; (**c**) pipe networks in study area.

**Table 1.** Land-use characteristics of the study area.

| Land Use | Area | |
|---|---|---|
| | (ha) | (%) |
| Water | 68.14 | 3.15 |
| Road | 698.34 | 32.32 |
| Roofs | 301.21 | 13.94 |
| Parking | 20.09 | 0.93 |
| Farmland | 295.87 | 13.69 |
| Greenspace | 777.12 | 35.96 |

*2.2. Data Preparation*

In this study, three types of data were collected: (1) physical data, including land use, digital elevation model (DEM), and soil properties; (2) hydraulic data, including drainage system, outfall distribution, and river cross-section topographic data; and (3) hydrological data: observed rainfall–runoff data, the flow and water level at the three flow cross-sections M1, M2, and M3, as shown in Figure 3. Infiltration and rainfall characteristics were also analyzed in this study.

### 2.2.1. Infiltration Characteristics

The infiltration loss can be calculated using the following Horton equation:

$$f = f_c + (f_0 - f_c)e^{-kt} \tag{1}$$

where $f$ is the infiltration rate, $f_c$ is the final constant infiltration rate, $f_0$ is the initial infiltration rate, $k$ is an empirical constant which is related to the soil properties, and $t$ is the elapsed time.

To determine the parameters in Equation (1), a soil infiltration experiment was carried out in the study area. According to the distribution of the soil type, a total of 11 soil infiltration experiment sites were selected (Figure 3) and the Global Positioning System (GPS), which was produced by Garmin company of the United State, was used to record their coordinates and elevation. The infiltration characteristics were measured by the double loop method. Table 2 shows the results from parameterizing Equation (1) by fitting the curve with MATLAB of the United States MathWorks company using the measured infiltration data from 11 sites.

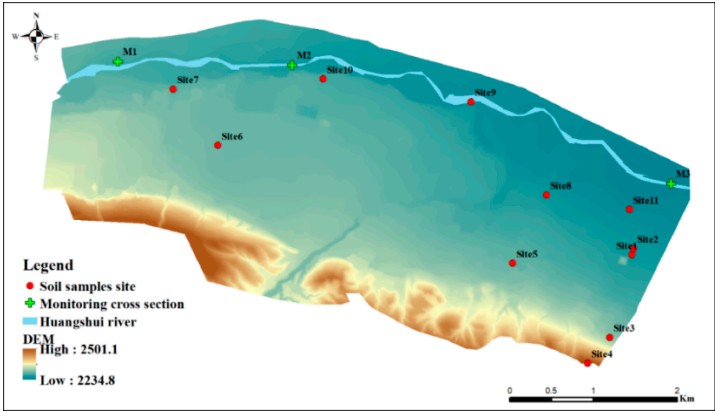

**Figure 3.** Locations of soil sample sites and cross-sections of Huangshui River.

**Table 2.** The parameters of Horton equation in the study area.

| Sites | $f_0$ (cm/min) | $f_c$ (cm/min) | $k$ |
|---|---|---|---|
| Site1 | 9.61 | 0.196 | 0.045 |
| Site2 | 4.92 | 1.604 | 0.047 |
| Site3 | 3.25 | 0.518 | 0.468 |
| Site4 | 7.03 | 0.530 | 0.232 |
| Site5 | 10.51 | 0.732 | 0.412 |
| Site6 | 15.56 | 1.476 | 0.114 |
| Site7 | 9.98 | 1.230 | 1.070 |
| Site8 | 19.69 | 1.539 | 0.229 |
| Site9 | 7.23 | 1.176 | 1.367 |
| Site10 | 7.04 | 0.633 | 0.486 |
| Site11 | 4.15 | 1.596 | 0.130 |

### 2.2.2. Rainfall Characteristics

In this study, to better understand the efficiency of the LID controls on the stormwater in the study area, three different return periods (T = 2, 5, and 10 years) were used in the analysis. The following rainfall intensity–duration formula in Xining was used to describe the rainfall characteristics in the study area:

$$q = \frac{461.9(1 + 0.9931\lg T)}{(t+3)^{0.686}} \tag{2}$$

where $q$ is the rainfall intensity (l/s ha), $t$ is the rainfall duration (min), and $T$ is the return period (year).

According to the nine years of rainfall data (2010–2018) obtained from the Xining Meteorological Bureau, the rainfall in Xining was dominated by the single striker type, and the peak rainfall intensity mostly occurred around r = 0.375, where r is the time to peak ratio, with a range of 0 < r < 1. Therefore, in this study, the Chicago storm profile was used to develop the design rainfall hyetographs at the study area. For the design rainfall hyetographs, the rainfall duration was 120 min and the time interval was 10 min. Figure 4 shows the design rainfall hyetographs for the three return periods.

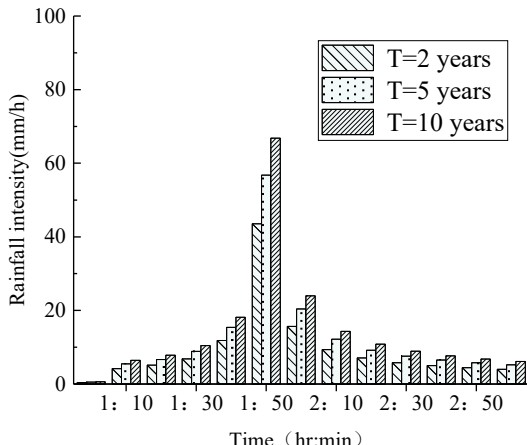

**Figure 4.** Design rainfall hyetographs at the study area.

### 2.3. Hydrologic and Hydrodynamic Models

### 2.3.1. Models Description

SWMM, developed by the United States Environment Protection Agency [31], was selected to simulate the rainfall–runoff process. The SWMM is a hydrodynamic-based rainfall–runoff and water quality simulation model. The runoff module comprehensively treats the processes of precipitation, runoff, and pollution load occurring in each subcatchment. The sink flow module is transported through the facilities such as the pipe network, channels, water storage and treatment facilities, water pumps, and regulating gates. The model can track the water quality and quantity of runoff generated by each subcatchment at any time, as well as the flow, water depth, and water quality of each pipe and channel. Since its development in 1971, SWMM has undergone several upgrades, and has been widely used around the world for planning, analysis, and design of stormwater, confluent sewers, sewage pipelines, and LID controls [32–35]. In the model, LID control can be represented by a combination of vertical layers whose properties include thickness and hydrological conductivity. Further, it can be placed in any subcatchment with any designed size [8].

MIKE 11, developed by the Danish Hydraulic Institute (DHI), was selected to simulate the excess stormwater discharge into rivers. MIKE 11 is based on the complete dynamic wave formulation of the Saint-Venant equations and can represent a wide range of hydraulic structures such as weirs, gates, and bridges [19]. It uses the finite difference approach and "double sweep" algorithm to solve the unsteady state flow equations [18]. The model can be used to simulate various vertical and homogeneous flow conditions from steep streams to tidal estuaries. In addition, the model can simulate various simplified flows.

In this study, based on land-use type and the distribution of the drainage system, 272 subcatchments, 310 junctions, 311 conduits, and 17 outlets were used to model the study area, as shown in Figure 5. Three LID source controls, vegetated swale (VS), permeable pavement (PP), and rain garden (RG), were applied within the urban catchment. According to the SWMM 5.0 manual and some relevant literature [8,10,31], the LID controls parameters were set to the values as shown in Table 3. The Horton equation was used to estimate the infiltration loss and the surface runoff was calculated using Manning's equation. The dynamic wave theory was used to calculate the flow routing. The volume of water from

the pipe network directly discharged into the Huangshui River was simulated and MIKE 11 was used to describe water flow evolution in the river (Figure 3). The variation of flow in the Huangshui River was used to evaluate the effectiveness of LID controls under the different spatial distributions with reduced DCIA at the urban catchment scale.

**Table 3.** Values of LID controls parameters.

| LID Control | | Parameters | | | | |
|---|---|---|---|---|---|---|
| Permeable Pavement (PP) | Surface | Berm height (mm) | Vegetation (%) | Manning's n | Surface slope (%) | |
| | | 5 | 0 | 0.013 | 1 | |
| | Pavement | Thickness (mm) | Void (%) | Imperviousness (%) | Conductivity (mm/h) | Clogging factor |
| | | 150 | 15 | 30 | 72 | 100 |
| | Storage | Thickness (mm) | Void (%) | Conductivity (mm/h) | Clogging factor | |
| | | 180 | 75 | 76 | 100 | |
| Vegetated Swale (VS) | Surface | Berm height (mm) | Vegetation (%) | Manning's n | Surface slope (%) | Side slope (%) |
| | | 300 | 90 | 0.15 | 1 | 33.3 |
| Rain Garden (RG) | Surface | Berm height (mm) | Vegetation (%) | Manning's n | Surface slope( %) | |
| | | 250 | 90 | 0.1 | 1 | |
| | Soil | Thickness (mm) | Porosity | Conductivity Slope | Conductivity (mm/h) | Suction head (mm) |
| | | 100 | 0.5 | 10 | 72 | 38 |

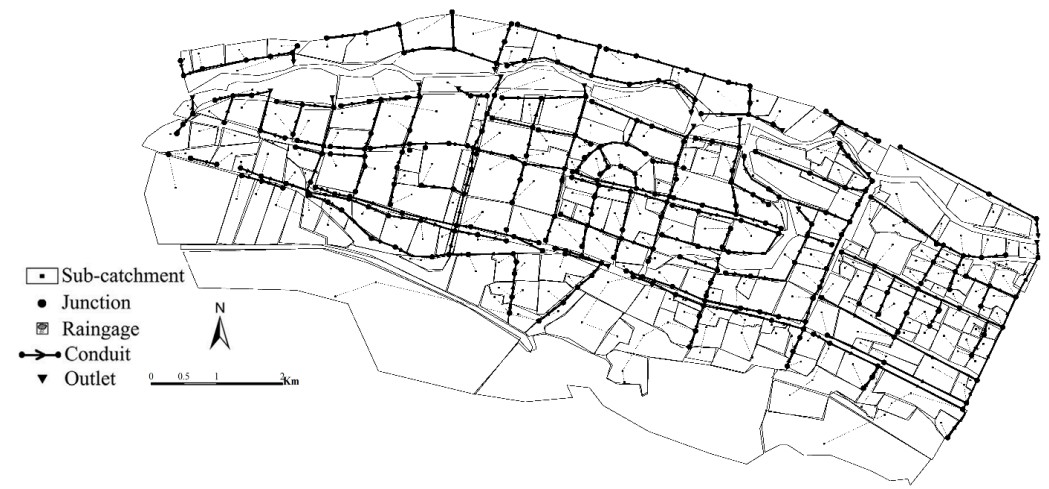

**Figure 5.** Hydrological model of study area.

### 2.3.2. Model Calibration and Validation

The model calibration and validation were based on comparisons of the observed and simulated runoff processes and changes in the water level. In order to assess the model performance, the Nash–Sutcliffe efficiency index (NSE) and $R^2$ were used as the goodness-of-fit indexes. Generally, the calibration model is considered satisfactory if the NSE and $R^2$ are greater or equal to 0.5 [36–38].

Using six selected rainfall events between September and October 2017, the SWMM model was calibrated and validated. Three rainfall events between 25 September and 4 October 2017 were used for the model calibration and the other three rainfall events between 5 and 10 October 2017 were used for the model validation. The quantitative assessment of the SWMM model performance is shown in Figure 6. For the model calibration, the comparison of the observed and simulated runoff hydrographs shows that NSE = 0.69 and $R^2$ = 0.75. The observed and simulated times to peak were exactly the

same, and the peak flows were almost the same. For the model validation, NSE = 0.51 and $R^2$ = 0.72. These results show that the SWMM model is suitable for simulating the hydrologic response in the study area.

MIKE 11 was calibrated by changing Manning's roughness coefficients, n, and comparing the simulated water level with the observed water level. As shown in Figure 7, the observed water levels at the monitoring cross-sections M2 and M3 were compared with the corresponding simulated water levels. At M2, NSE = 0.51 and $R^2$ = 0.71. At M3, NSE = 0.52 and $R^2$ = 0.77. These results show that MIKE 11 is suitable for simulating the variations of the water levels in the Huangshui River.

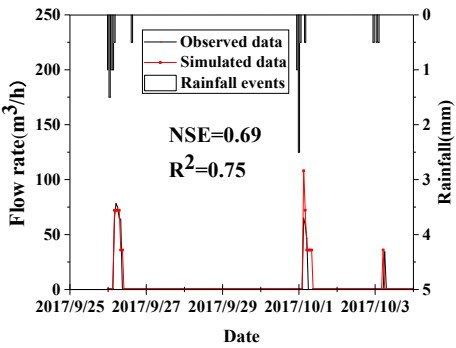 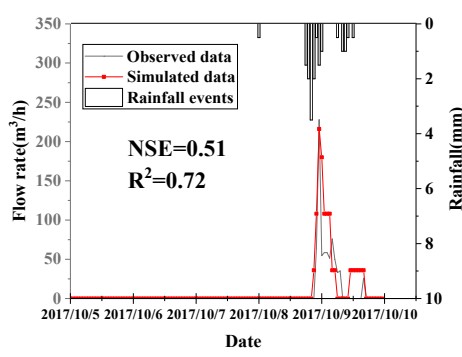

(**a**) Model calibration from 2017/9/25 to 2017/10/4     (**b**) Model validation from 2017/10/5 to 2017/10/10

**Figure 6.** Comparisons between observed and simulated runoff data.

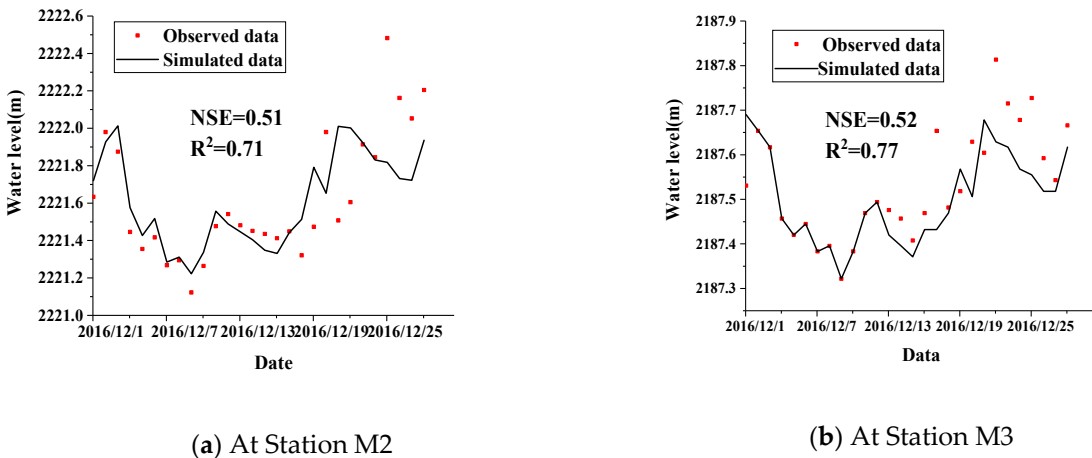

(**a**) At Station M2            (**b**) At Station M3

**Figure 7.** Comparisons between observed and simulated water levels at Stations M2 and M3 in the Huangshui River.

### 2.4. Identification of Directly Connected Impervious Area

In this study, GIS was used to calculate and identify the spatial distribution of DCIA. First, based on the land-use type, a comprehensive impervious surface layer, such as roads, residences, and pavements, were created. Then, based on the characteristics of each subcatchment area, geospatial and hydrological information of the subcatchment and its properties were assigned. Further, using the Location Selection tool in ArcGIS, all of the impervious surface layers of the catchment were intersected with the drainage network system. Finally, based on the drainage networks, impervious areas with the characteristics of the subcatchment that were considered as DCIA were identified [10,26,27].

## 2.5. Design Scenarios

Based on the distribution of the DCIA in the study area, five design scenarios were developed. There are three routes for overland flow in the SWMM: outlet, impervious surface, and pervious surface. The subcatchment with the overland flow path as the outlet or impervious surface was changed to pervious surface with the implementation of the LID controls, thus reducing the DCIA (Figure 8). Scenarios 1–5 had the same area LID controls implemented in the study area but at the different spatial distributions to reduce the DCIA. All the design scenarios are described in Table 4 and the spatial distributions of LID controls are shown in Figure 9.

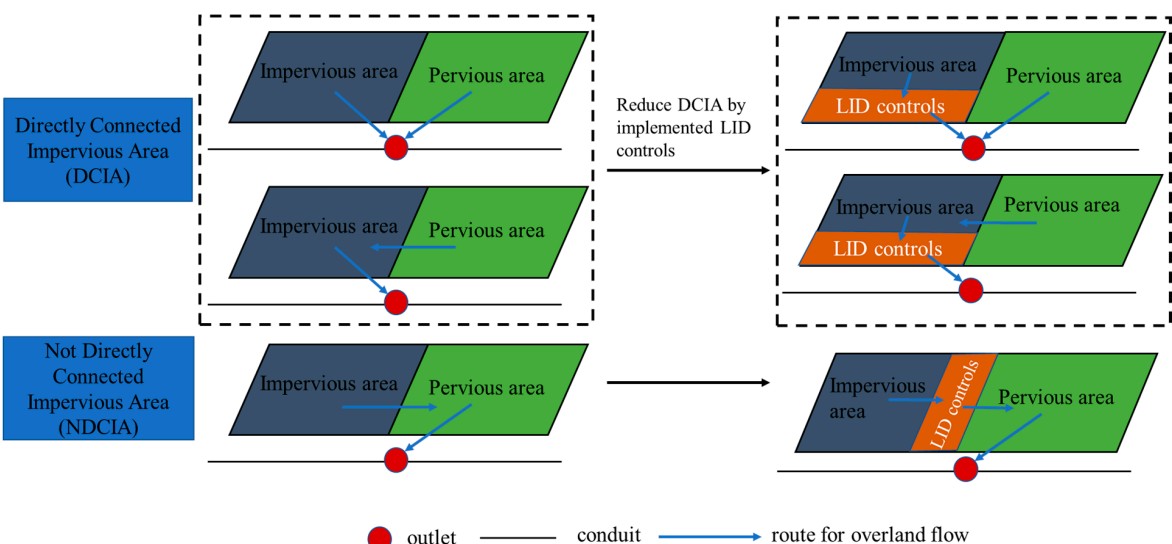

**Figure 8.** Reducing directly connected impervious area by implementing LID controls.

**Table 4.** Descriptions of five design scenarios.

| Scenario | LID Controls Applied | DCIA Reduction | Scenario Description |
|---|---|---|---|
| S1 | PP + VS + RG | 0% | All the LID controls were implemented in impervious areas that are not directly connected to each other. The storm runoff flows to streams via the non-directly connected impervious areas (NDCIA). So, the DCIA was not reduced. |
| S2 | PP + VS + RG | 25% | A small part of the LID controls were implemented in the DCIA in order to reduce the DCIA by 25%. The other LID controls were implemented in the NDCIA. |
| S3 | PP + VS + RG | 50% | Half of the LID controls were implemented in the DCIA in order to reduce the DCIA by 50%. The other LID controls were implemented in the NDCIA. |
| S4 | PP + VS + RG | 75% | Most of the LID controls were implemented in the DCIA in order to reduce the DCIA by 75%. The other LID controls were implemented in the NDCIA. |
| S5 | PP + VS + RG | 100% | All the LID controls were implemented in the DCIA, and this scenario disconnected all directly connected impervious areas. So, there was no DCIA in this scenario. |

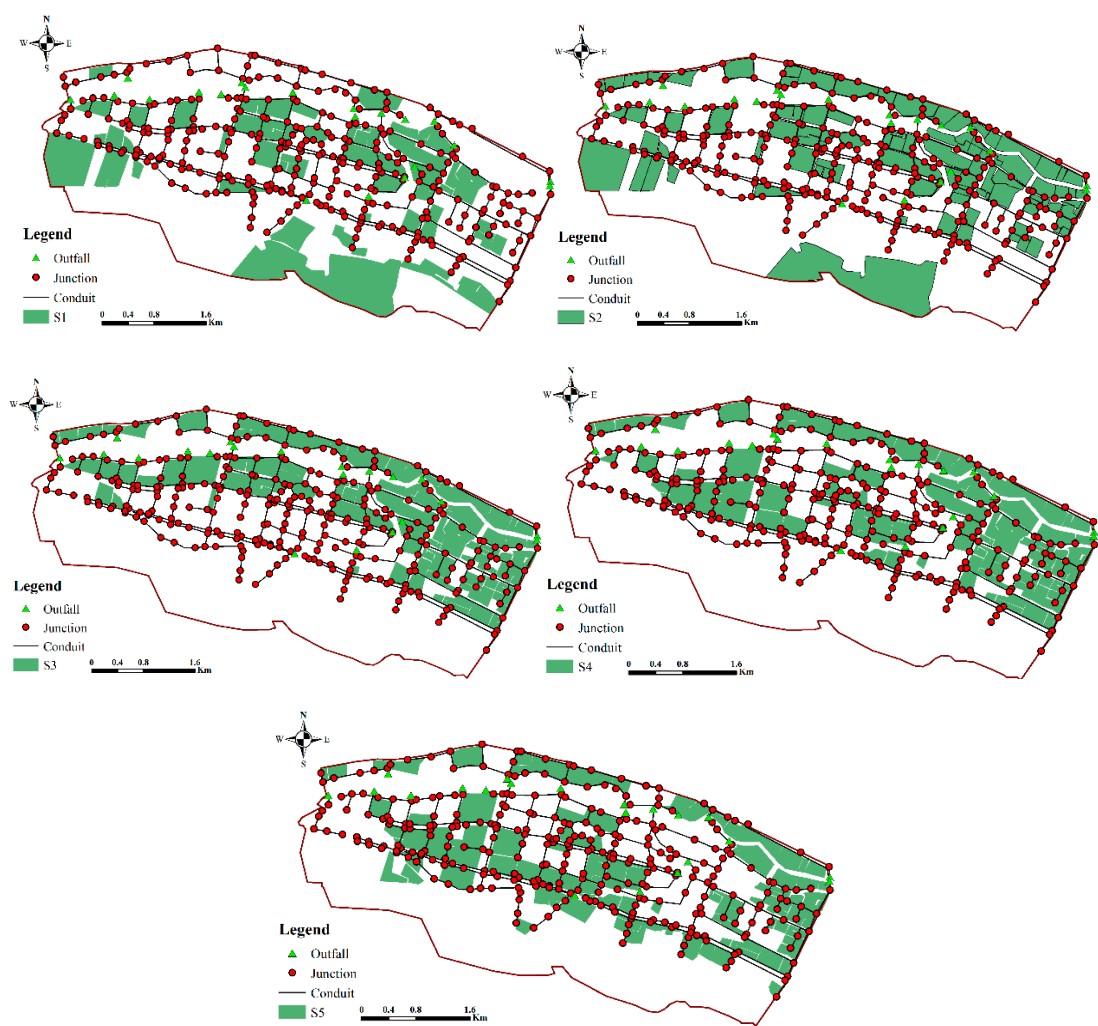

**Figure 9.** The spatial distributions of LID controls under five design scenarios.

## 3. Results

### 3.1. The Distribution of Directly Connected Impervious Area

DCIA is considered an important factor for investigating the efficiency of LID controls under different spatial distributions. Identifying the distribution of DCIA is important for effectively implementing LID controls.

As shown in Figure 10, the DCIA was distributed around the pipe network. According to the land-use type, the DCIA included almost all the areas of urban road and part of the pavements and roofs which were distributed in the commercial and residential districts. The not directly connected impervious areas (NDCIAs) were mainly distributed in the park and residential districts with great greenspace. According to the distribution of the DCIA, it was concluded that commercial and residential districts are the priority sites to implement the LID controls in the study area. This finding is consistent with Martin-Mikle et al. [39], who used the hydrologically sensitive areas as the impact factor and found that the buildings and roads were the priority sites under the local-scale LID placement.

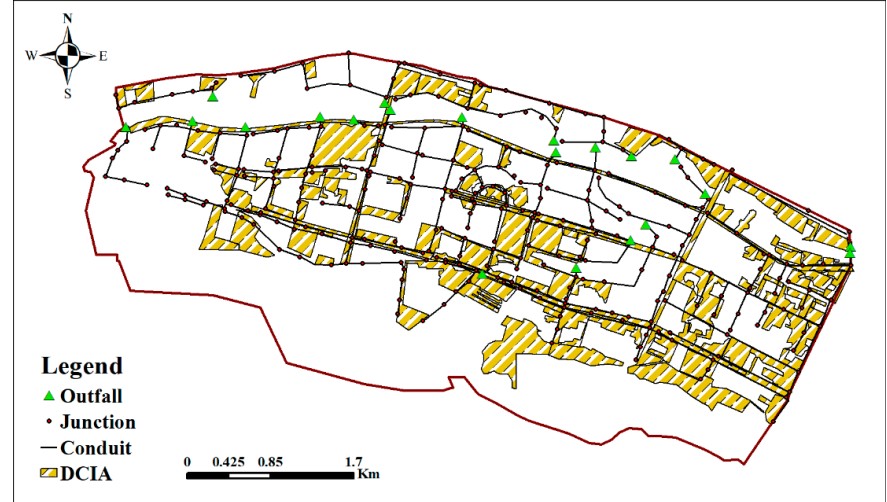

**Figure 10.** The distribution of directly connected impervious area within the study area.

### 3.2. Simulated Runoff under Five Design Scenarios

Figure 11 shows the simulated runoff changes resulting from the implementation of LID controls at different spatial distributions. The peak flow, runoff volume, and peak time for three design storms under five design scenarios are shown in Table 5. These results illustrate the relative effect of changes in DCIA on stormwater runoff. For example, for the design storm with T = 2 years, all five scenarios have the same rainfall amount, yet the hydrological performances are different. In S1, which had the largest DCIA in the study area, the peak flow, runoff volume, and peak time were 0.97 $m^3$/s, 25,648 $m^3$, and at 3:52, respectively. This scenario had the largest peak flow and runoff volume and the earliest peak time among the five design scenarios. From S2 to S5, the peak flow and runoff volume were gradually decreased and the peak time was gradually delayed. In S5, which had no DCIA in the study area, the peak flow, runoff volume, and peak time were 0.24 $m^3$/s, 7600 $m^3$, and at 4:02, respectively, which were the smallest values for peak flow and runoff volume and the latest peak time among the five design scenarios. These results confirm the effectiveness of sponge city construction in reducing the stormwater runoff and indicate that the efficiency of LID controls can be improved by DCIA reduction.

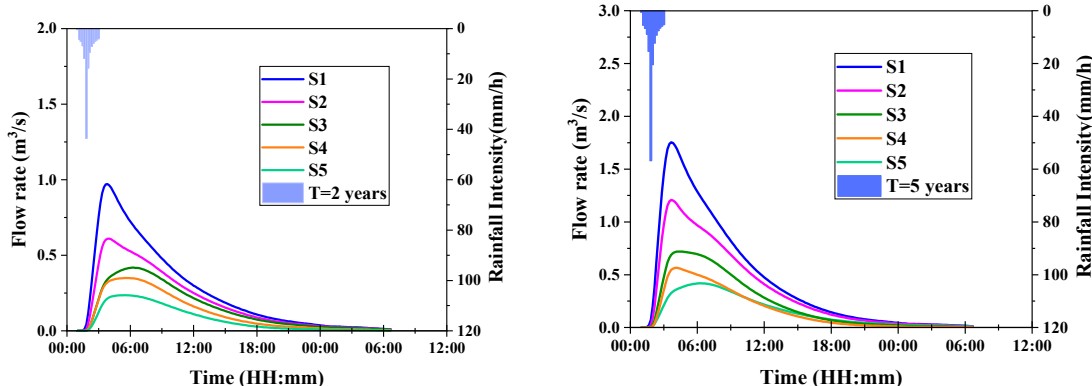

**Figure 11.** *Cont.*

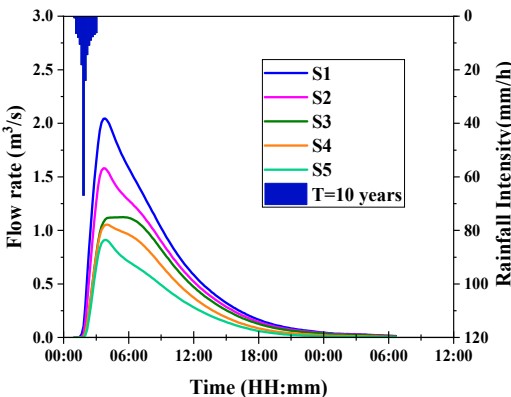

**Figure 11.** Simulated hydrographs for three storm return periods and five design scenarios.

**Table 5.** Hydrological performance for rainfall events of three return periods.

| Design Scenarios | Hydrological Performance | | | | | | | | |
|---|---|---|---|---|---|---|---|---|---|
| | Rainfall Event T = 2 years | | | Rainfall Event T = 5 years | | | Rainfall Event T = 10 years | | |
| | Peak Flow ($m^3$/s) | Runoff Volume ($m^3$) | Peak Flow Time (h:min) | Peak Flow ($m^3$/s) | Runoff Volume ($m^3$) | Peak Flow Time (h:min) | PEAK FLOW ($m^3$/s) | Runoff Volume ($m^3$) | Peak Flow Time (h:min) |
| S1 | 0.97 | 25,648 | 3:52 | 1.75 | 43,264 | 3:50 | 2.05 | 52,258 | 3:51 |
| S2 | 0.61 | 19,020 | 3:57 | 1.21 | 33,376 | 3:53 | 1.58 | 43,120 | 3:53 |
| S3 | 0.42 | 14,688 | 3:57 | 0.72 | 21,768 | 3:56 | 1.12 | 35,808 | 3:55 |
| S4 | 0.35 | 11,612 | 3:59 | 0.57 | 15,662 | 3:57 | 1.05 | 29,950 | 3:56 |
| S5 | 0.24 | 7600 | 4:02 | 0.42 | 14,688 | 4:01 | 0.91 | 22,790 | 4:00 |

### *3.3. Sensitivity of Hydrological Performance to DCIA Reduction*

The hydrological responses to the LID controls at four different spatial distributions (Scenarios 2, 3, 4, and 5) were assessed by comparison with Scenario 1, which had no DCIA. Figure 12 shows the changes in the peak flow reduction, runoff volume reduction, and peak time delay associated with changes in DCIA for the three rainfall events. Focusing on the 2-year design storm, the results point out that the reduction in hydrological performance was positively correlated and linearly dependent on the DCIA reduction. Furthermore, the peak flow reduction was most sensitive to the changes in the DCIA reduction while the peak time was the least sensitive to changes in DCIA reduction. This phenomenon also applied to the 5-year and 10-year design storms. By comparing the simulated results for the three rainfall events, the regression lines for the 2-year and 5-year storms were steeper than that for the 10-year storm. This indicates that as the rainfall intensity increases, the hydrological performance becomes more sensitive to rainfall intensity than reduction in DCIA.

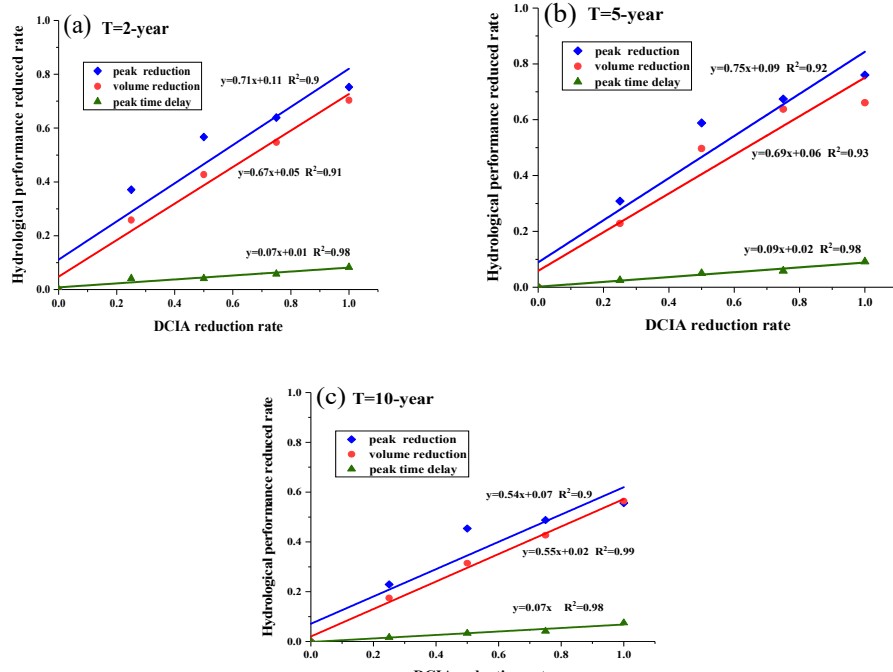

**Figure 12.** The effect of DCIA on hydrological performance for rainfall events of three return periods: (**a**) T = 2-yesr; (**b**) T = 5-year; (**c**) T = 10-year.

## 4. Discussion

### 4.1. The Effect of Directly Connected Impervious Area Reduction

The impervious area which was directly connected to the drainage network (DCIA) caused a higher peak flow and runoff volume and earlier peak time than the unconnected impervious area [22,26]. Areas can be disconnected by implementing green infrastructure [10,23]. The simulated results showed that with the same areas of LID implementation, the more disconnected the DCIA, the more effective the stormwater controls. For example, for the 2-year design storm, the peak flow was reduced by 37.1% when 25% of the DCIA was disconnected, and it could be reduced by 75.3% when the DCIA was completely disconnected. As shown in Figure 12, the hydrological performance reduction was positively correlated and linearly dependent on the DCIA reduction. The peak flow was the most sensitive parameter for the reduction in DCIA, followed by runoff volume and peak time. A similar linear relationship has been reported in some studies. By investigating the impact of land-use change expressed as either pavement of urban areas (increase in the fraction of imperviousness) or as unsealing and infiltration (decrease in the fraction of imperviousness), Kleidorfer et al. [40] showed that with increased runoff, the impact on land-use indicators is nearly linear. By comparing the different land-use conversion scenarios (i.e., EIA reduction), Palla and Gnecco [41] found that the hydrologic performance increased linearly with the EIA reduction percentage. These results provide a better understanding of the relationship between hydrological performance and land use.

### 4.2. The Effect of the Rainfall Event Return Period

By comparing the slopes of the hydrologic performance regression lines for the three rainfall events (Figure 12), it can be concluded that when the rainfall intensity was small, such as that in the T = 2-year or T = 5-year storms, hydrological performance was more sensitive to land use than rainfall intensity. With the increase in rainfall intensity, the role of land use was less significant and stormwater was mainly controlled by rainfall intensity. The reduction in the runoff is mainly governed by the retention capability of the catchment which is related to the properties of the LID controls, such as thickness,

porosity, and conductivity [8,41]. If the rainfall intensity increases, and the infiltration rate is smaller than the rainfall intensity, this can cause more runoff and the performance of the LID controls is limited. Similar results were found by other research about the performance of the LID controls associated with rainfall intensity. Qin et al. [8] concluded that LID may become less effective for large rainfall events by modeling the effects of rainfall patterns on the LID measures in an urbanizing catchment in Shenzhen, China using SWMM. Palla and Gnecco [42] demonstrated the influence of the rainfall event return period on the reduction of the peak and volume by implementing a high spatial resolution model. As the rainfall intensity increased, the reduction of hydrologic performance decreased.

## 5. Conclusions

The hydrological performance of five design scenarios under three different return periods (T = 2, 5, and 10 years) were simulated using GIS-based SWMM and MIKE 11 at urban catchment scale in Xining, China. Hydrological response to the effectiveness of sponge city construction was analyzed, as well as the impact of different LID control spatial distributions to reduce DCIA. This research integrated the concept of sponge city construction within urban land-use planning to manage stormwater and provided a technical framework that demonstrated that the GIS-based SWMM and MIKE 11 models can be used to evaluate the effectiveness of sponge city construction at urban catchment scale from the source, midway, and terminal. The main results are summarized as follows:

1.  By comparing the simulated results of S1–S5, the results confirmed the effectiveness of sponge city construction in reducing urban stormwater runoff and showed that even though all these design scenarios were implemented with the same area of LID controls, Scenario 5 was the most effective in stormwater management. This indicates that the efficiency of LID controls can be improved by changing the spatial distribution of the LID controls to reduce the DCIA.
2.  The hydrological performance reduction was positively correlated and linearly dependent on DCIA reduction. The peak flow was the most sensitive parameter to reduction in DCIA, followed by runoff volume and peak time
3.  By comparing the simulated results for the rainfalls of three return periods, the regression lines for 2-year and 5-year storms were steeper than that for the 10-year storm. This indicates that as the rainfall intensity increased, the hydrological performance was more sensitive to rainfall intensity than reduction in DCIA.

These results provide new insights for land-use planners and stormwater managers to implement LID more cost-effectively.

**Author Contributions:** Conceptualization, C.L. and X.Z.; Data curation, X.Z.; Formal analysis, X.Z. and J.X. (Jing Xu); Funding acquisition, X.Z.; Methodology, C.L.; Project administration, X.Z.; Resources, J.X. (Jun Xia); Software, C.L.; Supervision, X.Z. and J.X. (Jun Xia); Validation, J.X. (Jing Xu); Writing—original draft, C.L.; Writing—review & editing, J.X. (Jing Xu) and D.S. All authors have read and agreed to the published version of the manuscript.

**Funding:** This research was funded by the National Natural Science Foundation of China, grant number: No.41890823.

**Acknowledgments:** We thank the Wuhan Newfiber Optoelectronics Co., Ltd for their assistance in data collection and Joey Smith for language polish. We also express our gratitude to the reviewers and editors for their comments on the manuscript.

**Conflicts of Interest:** The authors declare no conflict of interest.

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
