# Peer review of "The Effect of Sponge City Construction for Reducing Directly Connected Impervious Areas on Hydrological Responses at the Urban Catchment Scale"

_water, doi:10.3390/w12041163_

Round 1

Reviewer 1 Report

Please, check that all abbreviations (like LID - Low Impact Development) are understandable

Reviewer 2 Report

It is interesting paper showing complex hydrological rainfall-runoff modeling to assess the efficiency of rain harvesting solutions on the city scale. Authors applying well know commercial software to study the effect of sponge city solutions used in Chinese city Xining.

Following issues should be addressed in the revised version of the paper:

  1. Rainfall duration - why only 120 min rainfall was analysed? Is it anyhow related to the time of concentration? In my opinion the other storm events should be also analysed as for example in here:
    Kolerski, T., & Kalinowska, D. (2019). Mathematical Modeling of Flood Management System in the City of Gdańsk, Oruński Stream Case Study. Acta Scientiarum Polonorum Formatio Circumiectus18, 63-74.
  2. Mike 11 calibration – the numerical results showed very well match with the observed data over the first 2 weeks of simulation, and after it is about 0,5 m bellow the observed data. Please explain the reason.
  3. Landuse – authors use very few classes of landuse and as far as developed areas it should be fine, for the ‘greenspace’ (covering nearly 36% of the basin) it should be distinguish between different vegetation types (forest, grassland, barrow, maybe others?). Please add some commend on that.

Minor problems:

Equations numbering - please make sure the numbers over the text are correct

Abbreviations: many occurring in the text are not properly described by the full name i.e. LID

Reviewer 3 Report

Article: The Effect of Sponge City Construction for Reducing Directly Connected Impervious Areas on Hydrological Responses at The Urban Catchment Scale

This article analyzes the different low impact development (LID) controls scenarios in the city of Xining (China) to minimize the runoff. The simulation of the runoff was done by using the SWMM and MIKE 11 tools. In addition, three different return periods were analyzed. The authors concluded that LID measures were less effective for large rainfall events. Furthermore, the results of the different analyzed scenarios were presented.  

General observations:

Please, describe the acronym for the first time that it is present in the manuscript.

Line 46: “effects of LID..” Does LID mean “low impact development”?

Line 83: “…. uncertainty of EIA estimates”. What is the mean of EIA?

Please do not use an acronym in a subtitle. Please, write the full name in the subtitles 2.4, 3.1 and 4.1

Abstract:

Please, describe the acronym for the first time that it is present in the abstract.

Line 12: “It includes LID at the…” low impact development?

Line 17: “SWMM and …” Storm Water Management Model?

Keywords:

Please do not use an acronym as a keyword.

Introduction

Figure 1. The graphic must be explained for itself. Please, include the acronym meaning in the graph or in the caption.

Materials and Methods

2.1 Study Area. Please include the name of the main river and its length in the study area.

Figure 2. Please, put the figure (a) over figures (b) and (c)

Line 125: Is the number of equation (1) (2) or 2.1. Please, correct this.

Figure 4. The graphs are very small. It is difficult to read. Please, increase the sizes of these graphs. Are S1, S2, .. S11 the analyzed sites? Please, indicate this in the caption of the graph.

Figure 5. Please increase the size of the letters of the graph.

Line 160: “The sink flow module is transported through the pipe network, channels, water storage and treatment facilities, water pumps, regulating gates, etc.” Please do not use etc. An alternative text can be: The sink flow module is transported through the facilities such as the pipe network, channels, water storage and treatment facilities, water pumps and regulating gates.

Line 179. Please use a space between the word and its acronym. “vegetated swale(VS)、permeable pavement(PP) and rain garden(RG), ..” and also change the symbol after swale (VS) by a comma.

Line 194: “…the calibration model is considered satisfactory if the NSE and R2 are greater or equal 0.5 [36]”. This statement is very important for the results of the manuscript. For that, please include more references to this topic.  Please also correct "equal 0.5" using the following text equal to 0.5.

Line 230. “There are three pathways overland flow can flow in the SWMM, …”. This sounds such as a cacophony. Please improve the redaction.

Figure 9. Please do not use acronyms in the caption of the graphs.

Results

Line 248: “According to the land use type, The DCIA …”. Please use the lowercase letter for the word “,The”

Line 253: “…, who used the HSAs as the impact factor …”. What is the meaning of the HSAs? Please describe before the acronym.

Figure 11. Please do not use an acronym in the caption of the graphs.

Lines 265 and 268, you wrote 3:52 and 4:02. What are the units of these values? Please, include.

Line 266: From S2 to S5, the peak flow and runoff volume decrease and the peak time is later. My question is: the peak time is later of what? It is not clear. Please rewrite the sentence.

Discussion

Line 296. Land use management, including LID, can reduce the 296 DCIA by implementing green infrastructure to disconnect the DCIA[10,23]. This text is not clear. Please rewrite.

Conclusion.

Line 336. Please, put a full stop at the end of the paragraph.

Reviewer 4 Report

The paper deals with a topic of interest. The abstract contains too many abreviations and details regarding the implementation and results. The main objectives should be better stressed out and unnnecessary details, as well as pointing out the used methods should be reduced.

Figure 1 is not referenced.

Title of figure one should be renamed.

A list of abreviations could help for a fluent reading.

Good experiments description and discussion of results.

In the conclusion it should be poited aout the originallity brought by the paper

Round 2

Reviewer 2 Report

No additional questions